# Assessment of a Diffuser-Augmented Hydrokinetic Turbine Designed for Harnessing the Flow Energy Downstream of Dams

**Jerson R. P. Vaz** [1,*,†] **, Adry K. F. de Lima** [1,†] **and Erb F. Lins** [2]

1　Graduate Program in Natural Resources Engineering, Institute of Technology, Federal University of Pará, Belém 66075-110, PA, Brazil; adry@ufpa.br
2　Academic Unit of Cabo de Santo Agostinho, Federal Rural University of Pernambuco, Cabo de Santo Agostinho 54518-430, PE, Brazil; erb.lins@ufrpe.br
*　Correspondence: jerson@ufpa.br
†　These authors contributed equally to this work.

**Abstract:** Harnessing the remaining energy downstream of dams has recently gained significant attention as the kinetic energy available in the water current is considerable. This work developed a novel study to quantify the energy gain downstream of dams using a horizontal-axis hydrokinetic turbine with a diffuser. The present assessment uses field data from the Tucuruí Dam, where a stream velocity of 2.35 m/s is the velocity at which the highest energy extraction can occur. In this case, a 3-bladed hydrokinetic turbine with a 10 m diameter, shrouded by a flanged conical diffuser, was simulated. Numerical modeling using computational fluid dynamics was carried out using the Reynolds averaged Navier–Stokes formulation with the $\kappa - \omega$ shear stress transport as the turbulence model. The results yield good agreement with experimental and theoretical data available in the literature. Moreover, the turbine power coefficient under the diffuser effect could increase by about 55% for a tip speed ratio of 5.4, and the power output increased by about 1.5 times when compared to the same turbine without a diffuser. Additionally, as there are no hydrokinetic turbines installed downstream of dams in the Amazon region, the present study is relevant as it explores the use of hydrokinetic turbines as an alternative for harnessing the turbined and verted flow from dams. This alternative may help avoid further environmental impacts caused by the need for structural extensions.

**Keywords:** hydro turbines; diffuser; computational fluid dynamics





## 1. Introduction

The use of hydro turbines has been widely investigated, as such technologies can harness kinetic energy from water currents with low environmental impact [1,2]. These turbines can be used with a diffuser, which is typically implemented around the hydro rotor to improve its power output, usually resulting in an increase of about two times. According to [3], hydro turbines with diffusers take advantage of the Venturi effect, reducing the fluid pressure downstream and increasing the axial velocity through the rotor.

Although the diffuser has been widely used on hydro turbines, this technology was originally designed to be applied to wind turbines [3,4]. Thus, the diffuser-augmented turbine theory is typically based on wind turbines, and further work based on this assumption needs to be taken into account. For example, Reference [5] developed a new approach to the aerodynamic optimization of a wind turbine with a diffuser based on an extension of the well-known blade element theory (BET) and a simple model for diffuser efficiency. Their work showed that there was a 35% increase in the turbine power coefficient compared to the turbine without a diffuser. Moreover, Reference [6] developed an innovative approach for the performance analysis of wind turbines with a diffuser based on BET, where a more

general semi-empirical one-dimensional analysis was performed by extending the Glauert correction to avoid high values of the axial induction factor. It is worth noting that in all of these citations, the models indicate that the flow velocity at the rotor plane significantly increases, making the turbine more efficient.

Regarding hydro turbines with diffusers, several studies have been conducted in the literature. Reference [7] developed studies on the effects of viscous loss, flow separation, and base pressure for a ducted tidal turbine using computational fluid dynamics (CFD), demonstrating that the base pressure effect can significantly enhance performance. Reference [8] conducted a numerical study to investigate the use of diffusers to enhance the performance and viability of hydro turbines, reporting that power can increase by a factor of 3.1. Most recently, Reference [9] conducted a study on the use of the remaining energy downstream of hydropower plants in rivers to assess the hydrokinetic potential. In their work, a 3-bladed hydro turbine with a 10 m diameter was used, which demonstrated the ability to generate 204 MWh in the period of 2008 to 2013.

This paper presents a study to quantify the energy gain downstream of dams using a diffuser-augmented hydrokinetic turbine to harness the remaining energy of the flow. The aim is to assess the energy gain by adding a diffuser around the turbine rotor and comparing it with field data obtained from the Tucuruí Dam. A CFD modeling approach is employed using the finite volume method. The BET model, extended to turbines with a diffuser [6], is used to evaluate the present study. The BET considers the diffuser speed-up ratio, as it behaves similar to a duct [10]. In this case, the flow around a 3-bladed hydrokinetic turbine with a 10 m diameter shrouded by a flanged conical diffuser is analyzed. Good agreement is found between the CFD and BET models, demonstrating that the turbine efficiency may increase by about 55% under the diffuser effect.

In order to provide an overview of the work, the remainder of this paper is organized as follows. Section 2 presents the numerical approach, which includes the geometrical and operational details of the hydrokinetic turbine system. In Section 3, data from the Tucuruí Dam are presented for analysis and comparison. Section 4 discusses the results of the study, including numerical validation to ensure coherent results. Additionally, comparisons between CFD and BET are made in the same section. Section 4 presents the conclusions of this study.

## 2. Numerical Approach

### 2.1. Hydrokinetic Turbine Configuration

A 3-bladed hydro turbine with a 10 m diameter, designed in the Energy and Environment Laboratory (EEL) of the University of Brasilia, is employed [11,12]. The turbine has a hub diameter of 1.2 m and a blade shape built using the NACA65$_3$-618 airfoil, as shown in Table 1. The chord and twist angle distribution of the turbine are depicted in Figure 1, and they were designed using the classical Glauert optimization, as further described in [13]. The main optimization equations are:

$$a' = (1 - 3a)/(4a - 1) \tag{1}$$

and

$$16a^3 - 24a^2 + \left[9 - 3x^2\right]a + x^2 - 1 = 0 \tag{2}$$

where $a$ and $a'$ are axial and tangential induction factors, respectively, and $x = \Omega r / V_0$ is the local speed ratio. Those parameters maximize the turbine power coefficient $C_P$. According to [5] and further mentioned in [14], if the angle of attack is higher than the stall angle, Equations (1) and (2) are not valid as the local drag on the blade becomes considerably high, which is ignored in the potential theory. Thus, it is important to state that the Glauert optimization procedure is only valid for $\lambda > 1$. Therefore, in this work, the lowest $\lambda$ simulated is 3.2.

**Table 1.** Design parameters used in the modeling.

| Parameters | Values |
|---|---|
| Turbine diameter | 10 m |
| Hub diameter | 1.2 m |
| Number of blades | 3 |
| Water velocity | 0.9–3 m/s |
| Water density | 997.0 kg/m$^3$ |
| Rotational speed | 8–33.92 RPM |
| Airfoil profile | NACA 65$_3$-618 |

**(a)**

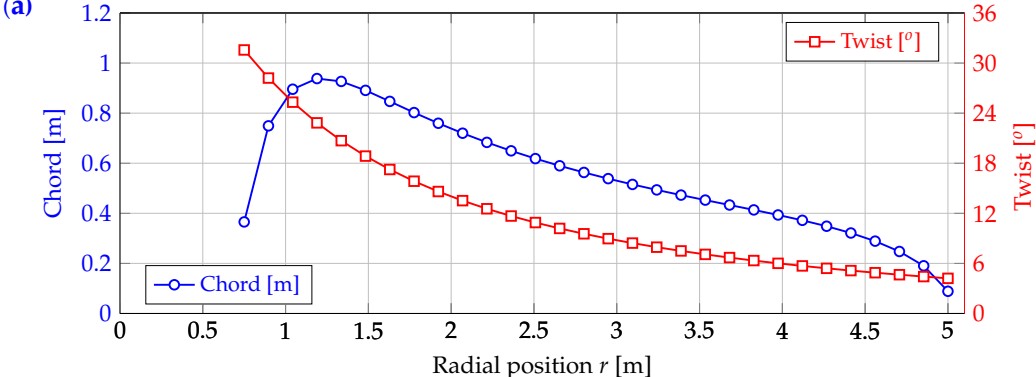

**(b)**

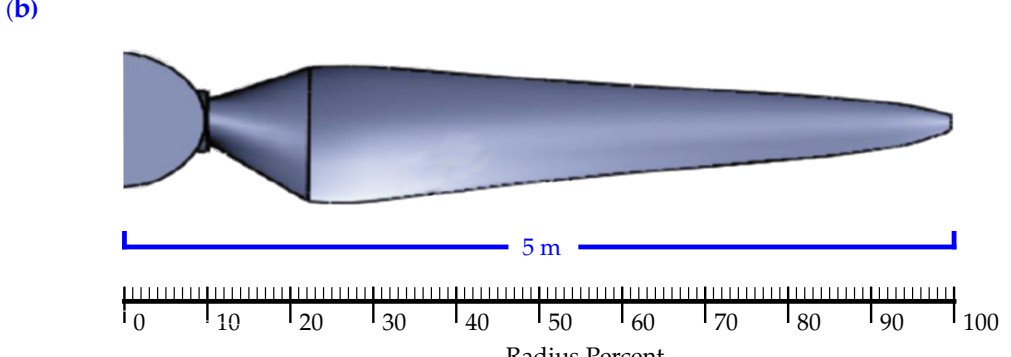

**Figure 1.** (**a**) Chord and twist angle distribution. (**b**) Blade shape.

The flanged conical diffuser designed by [11] is employed in this study. Figure 2 illustrates the diffuser geometry, which is based on the work of [15,16]. It has a cylindrical nozzle with a diameter of 1.01$D$ and a length of 0.14$D$, as well as a conical part with an opening angle of 4°. According to [15], the design concept for a turbine with a flanged diffuser is considerably different from that for a normal one, in which the local loading coefficient for the best performance of a flanged diffuser is considerably smaller than that for a turbine without a diffuser. Their investigation suggests that a relatively small loading coefficient, avoiding separation and maintaining a high pressure-recovery coefficient, tends to give high performance for a turbine with a flanged diffuser. As the flanged conical diffuser is easier to construct compared to those using airfoils, it is presumed that this kind of diffuser is a good choice to be used in this hydrokinetic turbine.

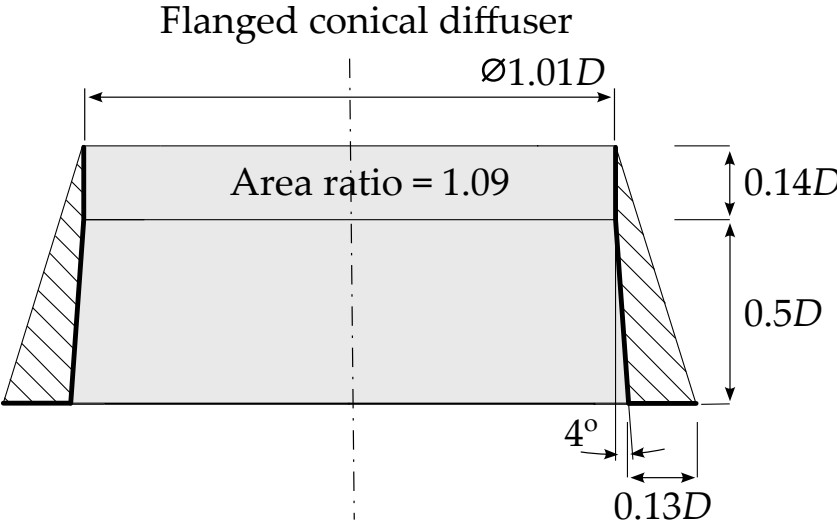

**Figure 2.** Illustration of the flanged conical diffuser (adapted from [3]).

### 2.2. Computational Modeling

The simulations were carried out through CFD modeling using the finite volume method. The mass conservation and Navier–Stokes equations were numerically approximated to obtain velocity and pressure fields. The flow was considered incompressible and turbulent. The effect of turbulence in the main flow was considered by using the Reynolds averaged Navier–Stokes (RANS) equations. In this methodology, the following decomposition is used [17]:

$$\mathbf{u} = \overline{\mathbf{u}} + \mathbf{u}' \tag{3}$$

$$p = \overline{p} + p' \tag{4}$$

where $\overline{\mathbf{u}}$ and $\overline{p}$ are the mean component of flow velocity vector $\mathbf{u}$ and pressure $p$, respectively, and $\mathbf{u}'$ and $p'$ are the fluctuation components of the variables. Therefore, the time-averaged form of mass and momentum conservation equations of the flow are given by

$$\nabla \cdot \overline{\mathbf{u}} = 0, \tag{5}$$

$$\frac{\partial \overline{\mathbf{u}}}{\partial t} + \overline{\mathbf{u}} \cdot \nabla \overline{\mathbf{u}} = -\frac{1}{\rho} \nabla \overline{p} + \nabla \cdot \left(2\nu \mathbf{S} + \frac{1}{\rho}\mathbf{R}\right) + \mathbf{f} \tag{6}$$

where $\rho$ is the density, $\nu$ is the kinematic viscosity, and $\mathbf{f}$ is the body force per unit of volume, which may represent Coriolis and centrifugal contributions. Moreover, $\mathbf{S} = \frac{1}{2}(\nabla \overline{\mathbf{u}} + \nabla^T \overline{\mathbf{u}})$. All terms in Equations (5) and (6) are functions of $\overline{\mathbf{u}}$ and $\overline{p}$. The only term where the velocity fluctuation has an impact is given by the Reynolds stress tensor $\mathbf{R} = -\rho \overline{\mathbf{u}' \times \mathbf{u}'}$. Since there is no explicit equation for this tensor, it should be modeled as a function of the mean parts of flow variables [18], usually by using a turbulence model. Many turbulence models are available in the literature, and this is still a very active research field (see, for example, [19–21]). In the present work, the shear stress transport (SST) turbulence model is applied [22]. In [23], this model was used to evaluate the flow around a three-bladed hydrokinetic turbine (without a diffuser), showing good results. In this model, the Reynolds stress tensor is computed as

$$\mathbf{R} = 2\rho \nu_T \mathbf{S} - \frac{2}{3}\rho \kappa \mathbf{I} \tag{7}$$

The $\kappa - \omega - \text{SST}$ is widely used in aeronautical flows as it can deal more accurately with strong adverse pressure gradients, boundary layer separation, transonic shock waves, etc. It combines the advantages of the Wilcox $\kappa - \omega$ model [19] in the vicinity of solid walls and the robustness of the $\kappa - \epsilon$ model in the regions of free-stream flows. This blending was first done in the work of [22] using the so-called baseline model (BSL). In the same work, a

second turbulence model is presented with the addition of a limiter of eddy viscosity, which can account for the transport of the turbulent shear stress. In that model, the turbulent viscosity and the transport equations for turbulent kinetic energy $\kappa$ and turbulent specific dissipation $\omega$ are given by:

$$\nu_T = \frac{a_1 \kappa}{\max(a_1 \omega, SF_2)} \tag{8}$$

$$\frac{\partial \kappa}{\partial t} + \mathbf{u} \cdot \nabla \kappa = P_\kappa - \beta^* \kappa \omega + \nabla \cdot [(\nu + \sigma_\kappa \nu_T) \nabla \kappa] \tag{9}$$

$$\frac{\partial \omega}{\partial t} + \mathbf{u} \cdot \nabla \omega = \alpha \frac{\omega}{\kappa} P_\kappa - \beta \omega^2 + \nabla \cdot [(\nu + \sigma_\omega \nu_T) \nabla \omega] + 2(1 - F_1)\sigma_{\omega 2} \frac{1}{\omega} \nabla \kappa \cdot \nabla \omega \tag{10}$$

The production term $P_\kappa$ and model functions $F_1$ and $F_2$ are

$$P_\kappa = \min(\mathbf{R} : \nabla \mathbf{u}, 10\beta^* \kappa \omega) \tag{11}$$

$$F_1 = \tanh\left(\left\{\min\left[\max\left(\frac{\sqrt{\kappa}}{\beta^* \omega y}, \frac{500\nu}{y^2\omega}\right), \frac{4\sigma_{\omega 2}\kappa}{CD_{\kappa\omega}y^2}\right]\right\}^4\right) \tag{12}$$

$$F_2 = \tanh\left[\left[\max\left(\frac{2\sqrt{\kappa}}{\beta^* \omega y}, \frac{500\nu}{y^2\omega}\right)\right]^2\right] \tag{13}$$

$$CD_{\kappa\omega} = \max\left(2\rho\sigma_{\omega 2}\frac{1}{\omega}\nabla \kappa \cdot \nabla \omega, 10^{-10}\right) \tag{14}$$

$$\phi = \phi_1 F_1 + \phi_2 (1 - F_1) \tag{15}$$

where $y$ is the distance to the nearest wall. The model constants $\alpha$, $\beta$, $\sigma_\kappa$ and $\sigma_\omega$ are blended using Equation (15) with values given by $\alpha_1 = 5/9$, $\alpha_2 = 0.44$, $\beta_1 = 3/40$, $\beta_2 = 0.0828$, $\sigma_{k1} = 0.85$, $\sigma_{k2} = 1$, $\sigma_{\omega 1} = 0.5$, $\sigma_{\omega 2} = 0.856$. Moreover, $\beta^* = 0.09$. More details about the model equations can be found in [24] and references therein.

The geometric model is defined by the sum of two volumes: a hexahedron and a disk, corresponding, respectively, to the static and rotating domains. However, it is worth mentioning that these two domains do not overlap, and the static domain and the rotating one are interconnected by an interface. According to [9], in order to parametrize and enable the simulation of turbines of varying sizes, the rotor radius must be used as an essential parameter for the construction of the control volume (static and rotating domains). The current dimensions of the domain as a function of the rotor radius $R$ can be seen in Figure 3.

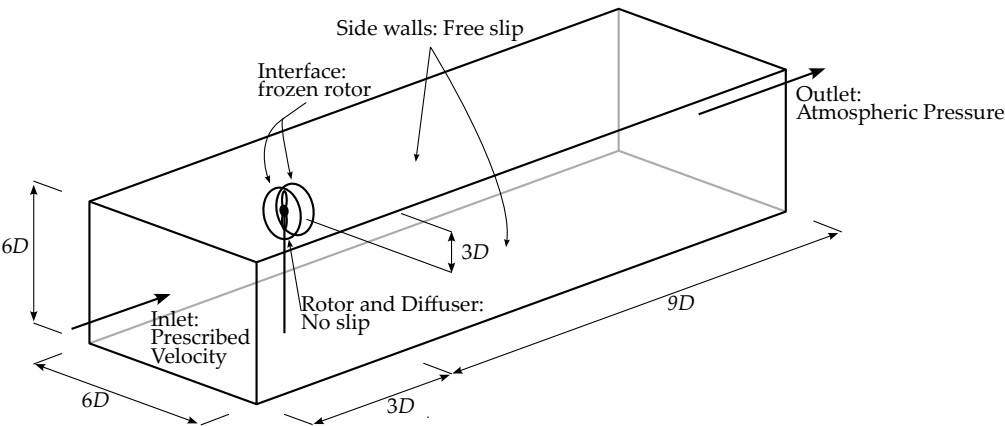

**Figure 3.** Setup of the computational domain.

The finite volume mesh is crucial since it will directly influence the velocity and pressure field quality. Therefore, to generate a suitable mesh for the geometric model, some relevant parameters are observed, such as $y^+$ and mesh skewness. $y^+$ is the dimensionless wall distance, defined by

$$y^+ = \frac{\rho y u^+}{\mu} \tag{16}$$

where $u^+$ is the wall shear velocity and $y$ is the dimensional distance to the wall. In this way, the near-wall region can be subdivided into three zones: the internal viscous layer ($y^+ < 5$), the buffer layer ($5 < y^+ < 30$), and the fully turbulent layer ($y^+ > 30$). These regions have a close resemblance to the turbulent log wall [25]. In order to correctly approximate the flow using the $\kappa - \omega - SST$ model, the maximum value of $y^+$ should be less than 1 [19]. Therefore, in order to meet this criterion, the inflation resource is used, which consists of a compilation of a number of prismatic layers over a certain surface. This leads to greater efficiency of the wall law and, consequently, correct computation of the pressure field variation in the region.

The skewness is also a dimensionless parameter that indicates the quality of the mesh, intended to verify cell geometric quality, which is the level of distortion suffered by a cell due to geometric proportions. The closer the skewness value is to 0, the greater the degree of geometric conformity and quality of the mesh, as measured by this dimensionless parameter [26]. These parameters are evaluated in this simulation.

Figure 4 shows the mesh distribution employed in the simulations. Due to the large computational domain required in this work, the mesh is refined in zones where the flow features most affect the turbine performance. The boundary conditions are shown in Figure 3, and they are set as follows. In the inlet zone, the velocity is prescribed with $V_0 = 2.35$ m/s. The lateral walls are set as walls with slip conditions (no tangential stress). As these walls are far from the turbine, this assumption is valid. In the outflow boundary, a static pressure specification is used. Between interfaces of static and rotating domains, the frozen rotor specification is employed to better determine the flow transition between domains. The boundary equations can be set as:

$$\mathbf{u} = \mathbf{U_0}, \frac{\partial p}{\partial n} = 0 \text{ at domain inlet} \tag{17}$$

$$\frac{\partial \mathbf{u}}{\partial n} = \mathbf{0}, p = 0, \text{ in domain outlet} \tag{18}$$

$$\mathbf{u} \cdot \mathbf{n} = 0, \frac{\partial \mathbf{u}}{\partial n} = 0, \frac{\partial p}{\partial n} = 0 \text{ in slip walls} \tag{19}$$

$$\mathbf{u} = \mathbf{0}, \frac{\partial p}{\partial n} = 0 \text{ in no-slip walls} \tag{20}$$

In these equations, $n$ represents the direction normal to the boundary surface, and $\mathbf{U_0}$ is the prescribed velocity vector. At the inlet, the turbulent intensity rate is set at 5%, while the turbulent viscosity ratio is $\nu_T / \nu = 10$. The present work uses the software Ansys CFX, with physical and numerical parameters, including fluid water at 25 °C, a static domain for the control volume and diffuser, and a rotary domain for the turbine rotor. A rotation speed of 26 rpm, a stream speed at the inlet of 2.35 m/s, and an average static pressure at the outlet of 0 Pa are employed. The residual type is the RMS, the advection scheme is set to high resolution, and the turbulence numerics are of the first order.

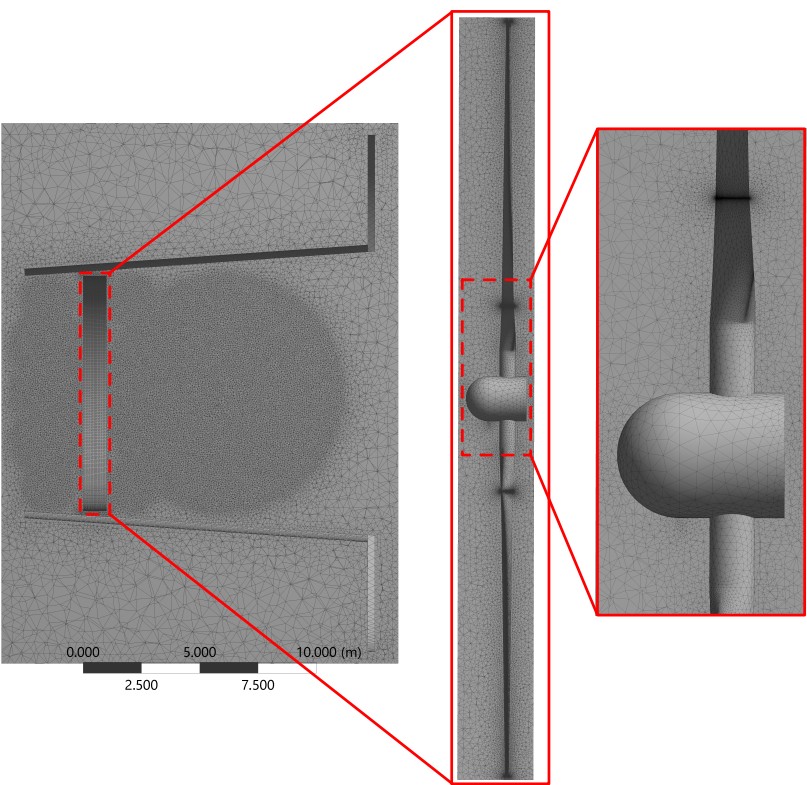

**Figure 4.** Mesh grid of the diffuser and rotor.

## 3. Remaining Energy Downstream of Dams

Usually, the remaining flow downstream of dams contains a significant amount of kinetic energy, making it worthy of attention. The goal of this study is to demonstrate that a hydrokinetic turbine with a diffuser can enhance the utilization of this kinetic energy. To achieve this objective, simulations were conducted for the flow downstream of the Tucuruí Dam, and a comparison is made between a hydrokinetic rotor with and without a diffuser. The Tucuruí Dam, located on the Tocantins River, is one of the largest hydroelectric power plants in the state of Pará. As shown in Figure 5, this dam consists of 25 Francis hydro turbines with a total installed capacity of 8370 MW, producing 21.4 TWh of electricity annually. It is the largest power plant in Brazil, as the Itaipu dam is a bi-national project between Brazil and Paraguay. The remaining energy downstream of the Tucuru'i Dam has recently been studied by [9], where velocities and depths were determined. Figure 6 shows the probability density distribution of the river stream velocity (where the average is $V_A = 1.63$ m/s) and the energy density. As reported in [9], the data consist of measurements collected from 01/01/2008 to 07/22/2013. The bathymetric data were provided by the Eletronorte/Eletrobrás Company. These data were obtained from the Administration of Hydroways of the Eastern Amazon (Administraç ao das hidrovias da Amazônia Oriental—AHIMOR) and were collected in September 2004 via a single-beam echo sounder. In this case, the highest energy occurs at $V_0 = 2.35$ m/s, which is used for designing the hydrokinetic turbine. The average depth of the river is 37.67 m, allowing for the use of a hydrokinetic turbine with a 10 m diameter. The depth effect on the turbine performance is not considered in this work. It is also important to highlight that in the Amazon region, the landscape competes with human occupations along the riverbeds, with family agriculture being the main means of economy. However, the lack of electricity is a common problem in the region. Thus, it is crucial to develop new studies to search for more efficient alternative designs to hydrokinetic turbines, aiming to present feasible methodologies that can be implemented in the Amazon context. Along the Tocantins River, many families still live without electricity. Typically, each residence has one electric iron, one fridge, one blender,

one 20-inch TV, one small fan, and about nine compact fluorescent light bulbs. This leads to an energy consumption of about 4.70 kWh per day.

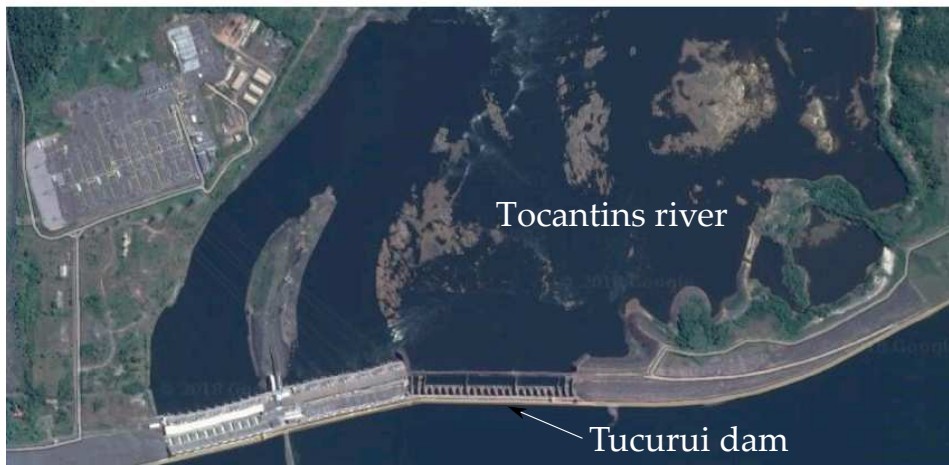

**Figure 5.** Satellite images of the Tucuruí Dam.

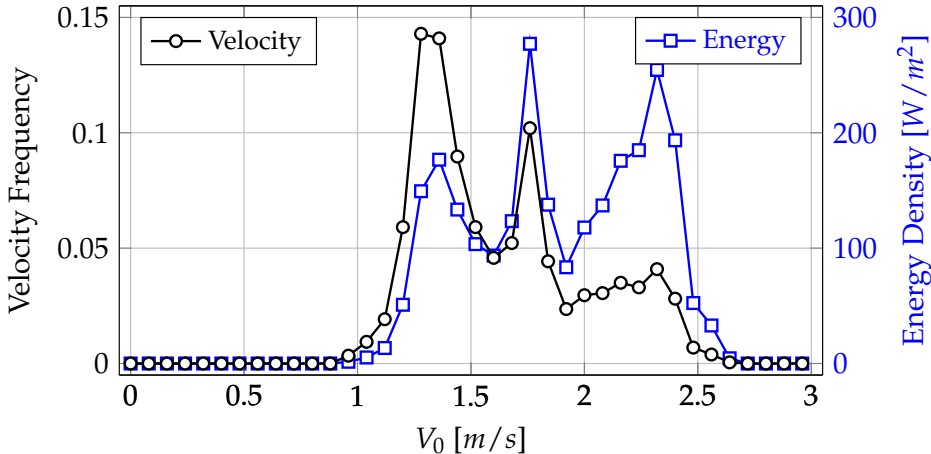

**Figure 6.** Velocity frequency and energy density of the river stream (adapted from [9]).

## 4. Results and Discussion

In this section, the results and discussions are presented as follows: Firstly, the numerical validation is presented by comparing the diffuser and rotor using experimental data. Secondly, a comparison is made using CFD modeling and the BET approach with variation in the water velocity, and the performance of the turbine is evaluated.

### 4.1. Numerical Validation

In this work, the numerical validation was carried out in two steps. First, the diffuser modeling was validated using the measurements made by [15]. This comparison is shown in Figures 7 and 8. Note that for an empty diffuser, the agreement between the present work and the experiments is good, showing physical consistency in the simulations. It is worth noting that the diffuser in Figure 7 is predominantly conical, while in Figure 8, it is conical with a flange at the outlet. These results demonstrate that the present validation is good enough to produce accurate results.

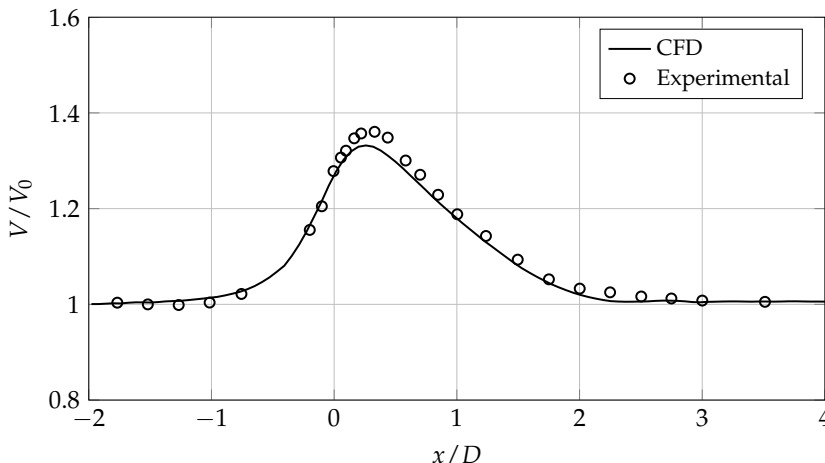

**Figure 7.** Flow velocity for a diffuser without flange.

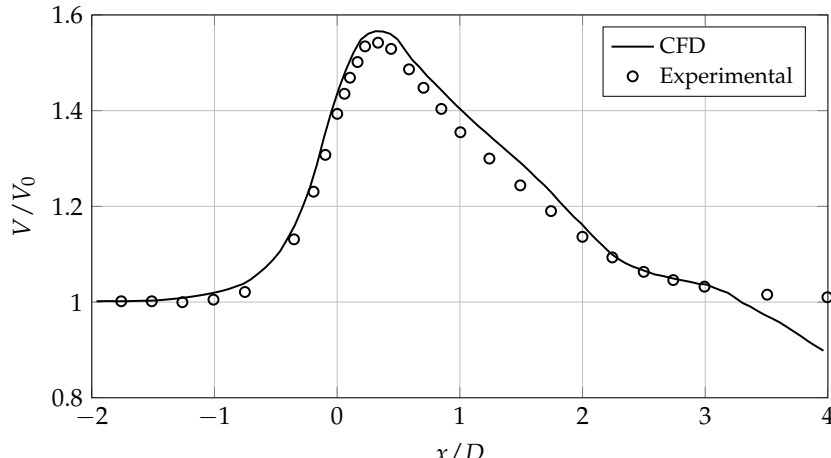

**Figure 8.** Flow velocity for a diffuser with flange ($h/D = 0.125$).

The second step of the validation uses the experimental results of the NREL Phase VI turbine from the National Renewable Energy Laboratory [27], with only the bare turbine considered. According to [11], the blade trailing edge and boundary layer are usually responsible for higher velocity gradients, requiring more effort regarding mesh refinement. Therefore, in the present work, to accurately model the boundary layer on the turbine blades, the region near the wall is divided into 25 prismatic element layers. This implementation is necessary for a detailed solution as the $\kappa - \omega - $ SST turbulence model is being used. Therefore, results for the mesh refinement study are shown in Table 2, in which the impact of the mesh refinement on the power output of the turbine is assessed for $\lambda = 5.4$. The table shows six different meshes developed for the rotor with different configurations for local refinement. The mean and maximum $y^+$ values at walls, as well as the power output, are presented. As the experimental power output is 6.1 kW, the error compared to the CFD modeling is 4.4% for Mesh 5, indicating good convergence of the numerical procedure. The mesh with $y^+ < 1$ and good geometric quality is Mesh 5, which has a skewness distribution of more than 90% of the cells with skewness of less than 0.5, and only a small percentage of cells (0.39%) with skewness greater than 0.75. In addition to the skewness, the mesh quality was also evaluated through the aspect ratio and orthogonality, whose results are 1.1613 and 0.9988, respectively, in the regions of interest (rotor and diffuser), demonstrating the good quality of the mesh. It is important to note that the validation considers the diffuser and turbine as separate entities due to the difficulty in finding detailed experimental data for a complete hydro turbine with a diffuser in the literature. Therefore, once accurate validations for the diffuser and rotor are completed, the proposed CFD modeling for the

entire system will be considered valid. Figure 9 shows the performance of the present validation, comparing the experimental data developed by [27] with CFD. The modeling shows good agreement, with even low $\lambda$ values well represented.

**Table 2.** Mesh refinement study.

| Mesh | Number of Nodes $[\times 10^6]$ | $y_{max}^+$ | $y_{avg}^+$ | Power $[kW]$ |
|---|---|---|---|---|
| Mesh 1 | 2.47 | 5.45 | 3.43 | 2.81 |
| Mesh 2 | 3.78 | 1.36 | 0.85 | 2.76 |
| Mesh 3 | 5.62 | 0.42 | 0.26 | 4.96 |
| Mesh 4 | 6.35 | 0.45 | 0.09 | 5.66 |
| Mesh 5 | 7.63 | 0.48 | 0.08 | 5.83 |
| Mesh 6 | 8.30 | 0.48 | 0.08 | 5.94 |

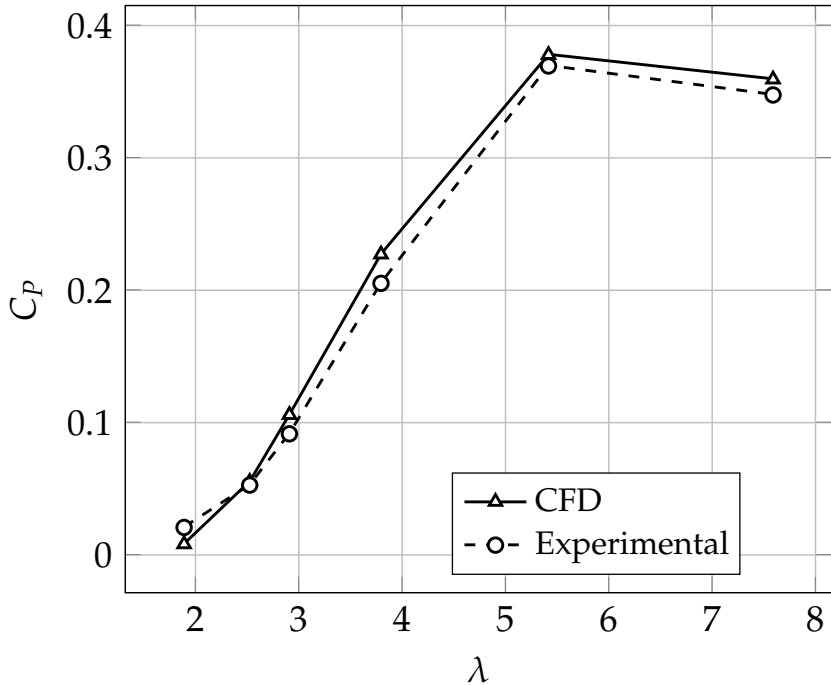

**Figure 9.** Output power as a function of $\lambda$—NREL Phase VI turbine [27].

*4.2. Performance of the Diffuser-Augmented Hydro Turbine*

To assess the performance of the hydro turbine, an extended BET model originally applied to wind turbines with the diffuser is used [6]. This model accounts for the effect of the diffuser through a parameter called the diffuser speed-up ratio, $\gamma$. In this work, the model is applied to a hydro turbine, which is valid as wind and hydro turbines operate on the same principles, as noted by [9,28,29]. In this section, the CFD results are compared to the BET model. As described by [6], the power coefficient is accounted for in the BET using the following expression for the axial induction factor $a$:

$$\frac{a}{1-a} = \frac{\gamma^2}{4} \frac{\sigma C_n}{\sin^2 \phi}. \tag{21}$$

where $\gamma$ is the diffuser speed-up ratio, $\sigma = Bc/2\pi r$ is the solidity, $\phi$ is the flow angle, and $C_n$ is the normal force coefficient. Figure 10 shows the velocity field throughout the turbine, revealing that the diffuser enhances the flow velocity at the rotor plane. At the blade tip, the water velocity is higher, leading to higher torque. However, it is essential to carefully examine high water velocity at the blade tip, as it can increase the risk of cavitation. The flow mass strongly increases under the diffuser effect, leading to an approximate 1.6 times

increase in stream velocity, as further elaborated on in Figure 8, where the rotor must be placed since it is the position with the highest water velocity along the turbine shaft. To evaluate the impact of the turbine on the velocity profile, Figure 11 depicts the velocity change caused by the rotor in the axial direction. As the rotor extracts kinetic energy from the flow, the stream velocity heavily decreases along the rotor center line due to the blockage effect imposed by the rotor on the diffuser-confined flow. According to [30], the blockage effect can significantly impact the rotor performance as it alters the flow around the blades, and it is a function of the number of blades [31].

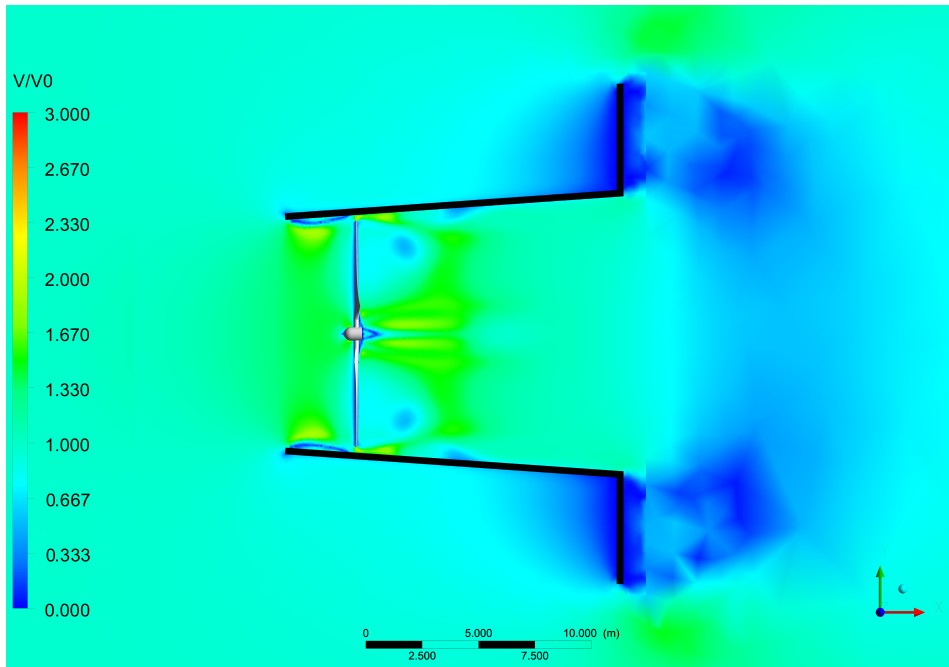

**Figure 10.** Velocity field throughout the hydro turbine.

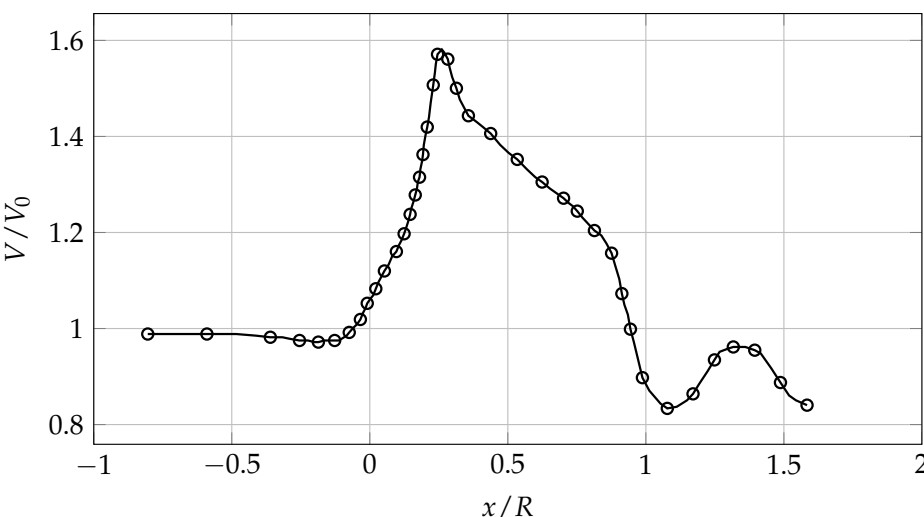

**Figure 11.** Flow velocity on the axial direction at the rotor center line.

The calculated turbulent kinetic energy (TKE) is shown in Figure 12 on a logarithmic scale. The turbulence levels are mostly in the range of $10^{-5}$ to $10^{-1}$ m$^2$/s$^2$. This result is consistent with previous studies, such as [23,32], which found higher TKE levels near the blade tips and at the back of the hub. Those studies analyzed free-flow turbines, where significant turbulence is generated at the blade tips. In the present work, the presence of

the diffuser greatly reduces the TKE levels, as the tip of the blade is very close to the inner wall of the diffuser, which energizes the boundary layer and delays its detachment.

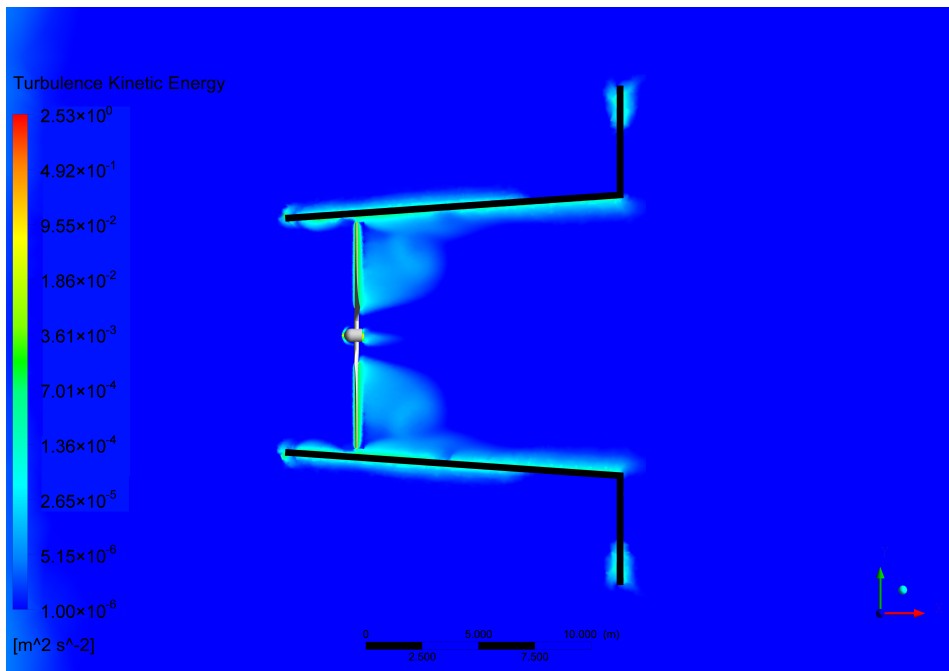

**Figure 12.** Turbulent kinetic energy at the rotor plane.

The output power of the turbine (with and without the diffuser) is shown in Figure 13. The agreement is good, even for a simplistic analysis, e.g., BET, which does not consider flow complexities, such as hydrodynamic interactions between the tip vortex and other turbulent structures, such as the boundary layer flow at the diffuser walls, high-pressure drop, and unsteadiness. Despite these assumptions, the present work exhibits relevant physical behavior. The output power of the turbine under the diffuser effect is approximately 45.5% higher compared to the turbine without the diffuser, for a stream velocity of 2.35 m/s. The turbine efficiency, represented by the power coefficient at $\lambda = 5.4$ using the diffuser, is 55% higher than a turbine without the diffuser, as demonstrated in Figure 14. It achieves 0.62, slightly exceeding the Betz–Joukowsky limit (0.59). This result is consistent with the findings of [29], where the increased power output of the shrouded hydrokinetic turbine is about 42% higher than a bare turbine for a water velocity of 2.5 m/s. However, economic feasibility should be evaluated, mainly for medium and larger hydro turbines, which involve larger structures. Regarding the power output for a stream velocity of 1.3 m/s in Figure 13, both turbines with and without the diffuser are close because the power output is dependent on $V_0^3$, resulting in similar values for stream velocities around 1 m/s.

In order to estimate the energy production of both turbines (with and without the diffuser) downstream of the Tucuruí Dam, Table 3 shows the results, considering the energy generated by the turbines between 2008 and 2013, as depicted in Figure 6. In this case, for a typical year, the turbine with the diffuser can produce 21.65 MWh more than the turbine without the diffuser. For a typical residence in the Amazon, the energy consumed is about 4.70 kWh per day, leading to 141.00 kWh per month. Thus, a turbine with a diffuser can produce energy for about 12 more residences than the turbine without a diffuser. This means that the diffuser technology may contribute to harnessing about 56% more energy, heavily improving the turbine performance. As recently demonstrated by [10], a diffuser produces a significant increase in the power coefficient, justifying the use of such technology to increase the turbine power output. Additionally, in the Amazon, there are no hydrokinetic turbines installed downstream of dams, making the present study relevant to the region, as the use of hydrokinetic energy is an alternative for harnessing turbined and

verted flows from dams. This alternative may prevent further environmental impacts on dams in need of structural extensions.

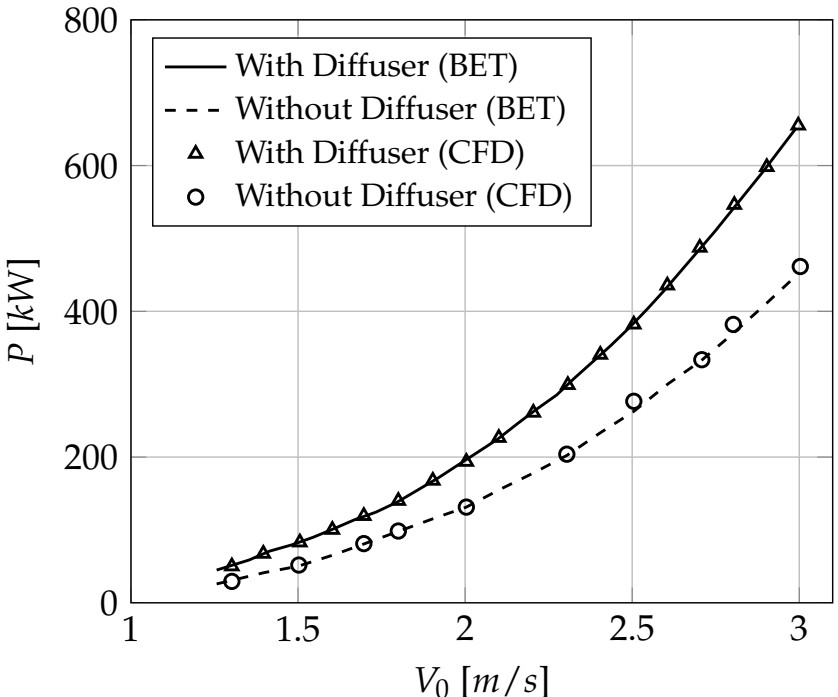

**Figure 13.** Power output as a function of the stream velocity.

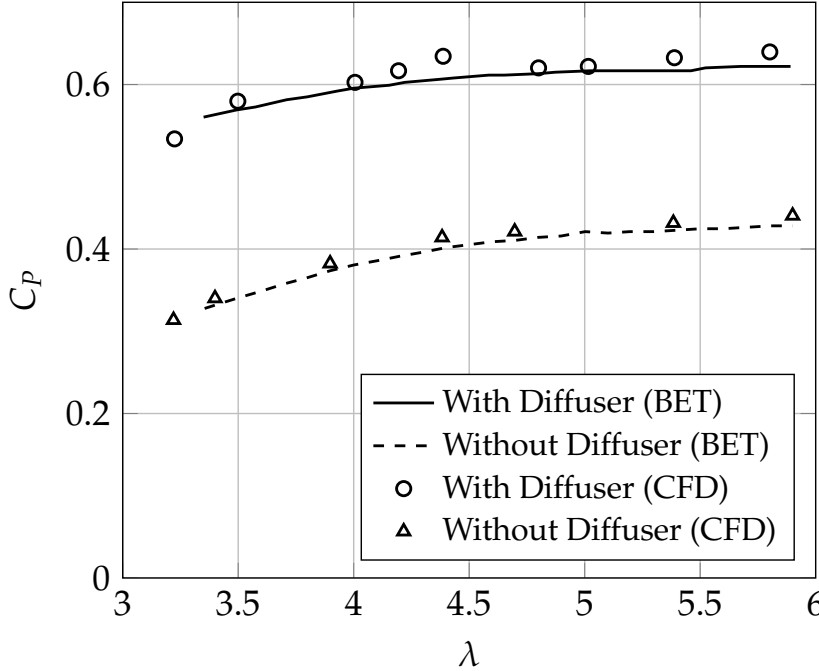

**Figure 14.** Power coefficient as a function of the tip speed ratio.

**Table 3.** Estimated energy.

| Period | Turbine Only [MWh] | Turbine plus Diffuser [MWh] |
|---|---|---|
| 2008 to 2013 | 216.34 | 337.23 |
| for a typical year | 38.75 | 60.40 |

## 5. Conclusions

This paper describes the energy gain of horizontal-axis hydrokinetic turbines under the diffuser effect using CFD simulations to assess the influence of the diffuser on water velocity at the turbine plane. The turbine power coefficient under the diffuser effect can increase by about 55% for a tip speed ratio of 5.4, and the power output can increase by up to 1.5 times compared to the same turbine without a diffuser. As the remaining energy downstream of dams usually has higher stream velocity, hydro turbines with diffusers can be good alternatives for harnessing even more kinetic energy. These alternatives may prevent further environmental impacts of dams in need of structural extensions.

It is necessary to consider some limitations of the present modeling, such as the lack of an analysis on the effect of the diffuser efficiency and thrust on the performance of the turbine. These two parameters can potentially affect the flow behavior through the rotor, as recently demonstrated by [33] and further detailed by [10,12,34].

Despite these limitations, the results obtained in this work are indeed relevant as diffuser technology significantly improves turbine efficiency and remains a good alternative, especially for supplying small communities downstream of dams, increasing the power output for 12 more families. As the Tocantins River is large, it can support more than 10 turbines with a diffuser, increasing the power supply for more than 120 families.

**Author Contributions:** Conceptualization: A.K.F.d.L., J.R.P.V. and E.F.L.; methodology: J.R.P.V.; software: A.K.F.d.L. and E.F.L.; validation: A.K.F.d.L., J.R.P.V. and E.F.L.; formal analysis: J.R.P.V. and E.F.L.; investigation: A.K.F.d.L.; resources: J.R.P.V. and E.F.L.; writing—original draft preparation: A.K.F.d.L., J.R.P.V. and E.F.L.; writing—review and editing: A.K.F.d.L., J.R.P.V. and E.F.L.; visualization: A.K.F.d.L.; supervision: J.R.P.V. and E.F.L.; project administration: J.R.P.V. All authors have read and agreed to the published version of the manuscript.

**Funding:** This research received no external funding.

**Institutional Review Board Statement:** Not applicable.

**Informed Consent Statement:** Not applicable.

**Data Availability Statement:** Not applicable.

**Acknowledgments:** The authors would like to thank the Brazilian National Council for Scientific and Technological Development (CNPq), the Coordination for the Improvement of Higher Education Personnel of the Brazilian government (CAPES), the Centrais Elétricas do Brasil (Eletronorte), and the Dean of Research and Graduate Studies of the Federal University of Pará—Brazil (PROPESP/UFPA) for the support.

**Conflicts of Interest:** The authors declare no conflict of interest.

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
