# Peer review of "Assessment of a Diffuser-Augmented Hydrokinetic Turbine Designed for Harnessing the Flow Energy Downstream of Dams"

_sustainability, doi:10.3390/su15097671_

Round 1

Reviewer 1 Report

Authors discuss analytical solution for CP but it has never been compare with CFD result. Then what is the purpose for reporting it (eqn 3).

Put the  math forms for boundary conditions – fig 4  does not suffice

Turbulence closure is not properly decribed, e.g. how turbulent viscosity is calculated. See for such description e.g.  10.1615/ComputThermalScien.2020035055 or https://doi.org/10.1115/1.4053547

Too much trivial matters are place in the  paper.

One sentence about skewnwess is enough – fig 10 is unnecessary.

Fig 6 and 7 can be combined by showing two ordinates in one plot. Authors should use space inteliigently.

Table  2 is never referred. Anyway it does not show grid independence properly as values are still varying with refine mesh.

Table 3 – include it in discussion otherwise what is the point in putting it.

Since you are performing RANS, show the difference in TKE for cxases with and without diffuser and link this with performance if possible.

Legends are not clear in fig. 12

Actually only few of the figures are referred  in the text (except Fig 12 shows..). They are not also discussed systematically.  That’s not the way an article is written. Please see the way a paper is to be written by observing published articles.

Author Response

Reviewer 1

  1. Authors discuss analytical solution for CP but it has never been compare with CFD result. Then what is the purpose for reporting it (eqn 3).

Answer: Thank you for your comment and suggestion. So, the power coefficient was compared with CFD as depicted in Fig. 16. On the Eq. 3, it was shown only for the purpose of including the optimization method used for the turbine blade. As suggested by the reviewer, the authors removed Eq. 3 from the paper.

  1. Put the math forms for boundary conditions – fig 4 does not suffice

Answer: Thanks for pointing it out. Following reviewer indications, Eqs. 8-11, were added.

  1. Turbulence closure is not properly described, e.g. how turbulent viscosity is calculated. See for such description e.g. 10.1615/ComputThermalScien.2020035055 or https://doi.org/10.1115/1.4053547

Answer: Following this suggestion, we have added the equations necessary to describe the turbulence model used.

  1. Too much trivial matters are place in the paper.

Answer: The authors have been reviewed the text, to make it a more fluid reading. The work develops a study to quantify the energy gain downstream of dams using horizontal-axis hydrokinetic turbine with diffuser in the Amazon. The assessment uses field data of Tucurui dam, which is in operation in the region. For this dam, the highest energy extraction can occur for a stream velocity of 2.35 m/s, from turbined and verted flow. In the region there are no hydrokinetic turbines installed downstream of dams, making the present study relevant, as it brings the use of hydrokinetic as an alternative for harnessing turbined and verted flow. This alternative may avoid further environmental impacts of dams in need for structural extension. For the authors, these facts are very important to the Amazon region, making this study relevant. To include this comment in the manuscript, it was added to the abstract, to the text right before Fig. 13, and the Conclusions, all in red color.

  1. One sentence about skewness is enough – fig 10 is unnecessary.

Answer: We agree with the reviewer on this point. We removed skewness figure and included a description about the skewness distribution in the text right after Fig. 8.

  1. Fig 6 and 7 can be combined by showing two ordinates in one plot. Authors should use space intelligently.

Answer: Thank you for this suggestion. Figure 6 was changed to include both data from figures 6 and 7.

  1. Table 2 is never referred. Anyway, it does not show grid independence properly as values are still varying with refine mesh.

Answer: The discussion and citation of this table is done in the paragraph just before the table. We highlight this information in red in the revised version.

  1. Table 3 – include it in discussion otherwise what is the point in putting it.

Answer: Thanks for the suggestion. This discussion was included in the manuscript, right before Fig. 13.

  1. Since you are performing RANS, show the difference in TKE for cases with and without diffuser and link this with performance if possible.

Answer: We have included a figure showing the Turbulent Kinetic energy in pages 11 and 12.

  1. Legends are not clear in fig. 12

Answer: Thanks for pointing this out. This figure was recreated for better legibility.

  1. Actually, only few of the figures are referred in the text (except Fig 12 shows). They are not also discussed systematically. That’s not the way an article is written. Please see the way a paper is to be written by observing published articles.

Answer: Thanks for your comment. Now, all figures are referred and discussed in the text. They are all highlighted in red.

Reviewer 2 Report

Some information about the experimental measurements must be provided. The authors must prepare a nomenclature and ca eliminate all the formula because they are well known for academics who work on CFD. Conclusions are not clear and the quantity of energy produced is not clear and appears low.

Author Response

Reviewer 2

  1. Some information about the experimental measurements must be provided. The authors must prepare a nomenclature and can eliminate all the formula because they are well known for academics who work on CFD. Conclusions are not clear and the quantity of energy produced is not clear and appears low.

Answer: Thanks for your comment. The experiments of the remaining energy of Tucuri dam are highlighted in page 7, and further details can be found in [10]. On the measurements of NREL Phase VI is detailed in [28], as pointed out on page 8.

Reviewer 3 Report

The manuscript is well written and can be of benefit for the scientific community. A few points to be considered to improve the manuscript quality:

- Minor language check for possible mistakes

- Discussing about mesh quality, skewness is not the only criteria which should be considered and discussed. other aspects such as non-orthogonality and aspect ratio are also rather important

- Please list the complete list of boundary conditions, e.g. turbulence

- Generally speaking, the manuscript is lacking a lot in the materials and methods section. Please consider adding all the needed information that the reader can reproduce the study, e.g. software used, full models, material properties, numerical schemes and solvers and solution settings

- Fig. 6: Vertical axis unit is missing

- Fig. 9&10: belong to the results section, as far as possible table and figures should be aligned with the text

- Table 2: It is not clear, which mesh was used? and the reason for selecting the mesh

- Fig. 12: It is not clear, the unit is missing and there is discontinuity in the fields at the flange section of the diffuser

- Fig. 15: the figures should be numbered based on their presence in the text

Author Response

Reviewer 3

To whom it may concern. The manuscript is well written and can be of benefit for the scientific community. A few points to be considered to improve the manuscript quality:

  1. Minor language check for possible mistakes

Answer: The authors really appreciate the comment of the reviewer. So, the paper has been thoroughly reviewed, and we have been improved many expressions.

  1. Discussing about mesh quality, skewness is not the only criteria which should be considered and discussed. Other aspects such as non-orthogonality and aspect ratio are also rather important.

Answer: We have checked these other parameters and the information was added to the manuscript right after Fig. 8, on page 9.

  1. Please list the complete list of boundary conditions, e.g. turbulence

Answer: Thanks for this suggestion, we have included this information on page 6 of the revised version, as well as a more detailed description of turbulence modelling used.

  1. Generally speaking, the manuscript is lacking a lot in the materials and methods section. Please consider adding all the needed information that the reader can reproduce the study, e.g. software used, full models, material properties, numerical schemes and solvers and solution settings

Answer: Thanks you for the suggestion. So, such a information was added in the text on page 6, right before Fig. 3.

  1. Fig. 6: Vertical axis unit is missing

Answer: Thanks for pointing this out. This is a graph of probability of density distribution. So, it does not have a unit in this axis. We have changed the description to make this clearer.

  1. Fig. 9&10: belong to the results section, as far as possible table and figures should be aligned with the text

Answer: We have corrected this in the revised version.

  1. Table 2: It is not clear, which mesh was used? and the reason for selecting the mesh

Answer: Thanks for noticing this. Table 2 shows the mesh convergence study. This table shows different mesh refinements, values of y+, and power calculated. The mesh selected for the analysis was number 5, because it presents a very good y+ value, in addition to a small difference in the value of power generated compared to mesh number 6, and has about 1.3 million fewer cells. The draft was lacking this information, so it was included in section 4.1.

  1. Fig. 12: It is not clear, the unit is missing and there is discontinuity in the fields at the flange section of the diffuser

Answer: This figure was recreated with more detail. Please note that, due to some figure removal, this is now figure 10 in the revised version.

  1. Fig. 15: the figures should be numbered based on their presence in the text

Answer: Thanks for noticing this error. We have corrected this issue in the revised version.

Reviewer 4 Report

This manuscript studied the assessment of a diffuser-augmented hydrokinetic turbine designed for harnessing the flow energy downstream of dams. The subject is interesting and the manuscript has been well written. The subject fits the scope of the journal and the novelties are desirable. Therefore, I recommend it for publication in the prestigious journal of "Sustainability" after addressing some minor points. 

1. The literature survey is good, up-to-date, and broad enough. The novelties and contributions of the study have been clearly highlighted in the last paragraph of the study. However, I recommend to start the last paragraph of the Introduction with an introductory sentence on the field.

2. The computational modeling section is perfect, reproducible, and well represented. Please double check all equations to be referenced where it is required.

3. Figure 12 is eye-catching and your effort for presenting this schematic is valuable. I recommend to show the velocity field throughout the turbine in 3D domain where the diffuser increases the flow velocity at the rotor plane. 

Author Response

Reviewer 4

This manuscript studied the assessment of a diffuser-augmented hydrokinetic turbine designed for harnessing the flow energy downstream of dams. The subject is interesting and the manuscript has been well written. The subject fits the scope of the journal and the novelties are desirable. Therefore, I recommend it for publication in the prestigious journal of "Sustainability" after addressing some minor points.

  1. The literature survey is good, up-to-date, and broad enough. The novelties and contributions of the study have been clearly highlighted in the last paragraph of the study. However, I recommend to start the last paragraph of the Introduction with an introductory sentence on the field.

Answer: Thank you for the suggestion. This was included in the last paragraph of the Introduction.

  1. The computational modeling section is perfect, reproducible, and well represented. Please double check all equations to be referenced where it is required.

Answer: Thank you for this comment. We have added more information about the turbulence model and boundary conditions, and more references about the numerical method. We think that this section is now more informative.

  1. Figure 12 is eye-catching and your effort for presenting this schematic is valuable. I recommend to show the velocity field throughout the turbine in 3D domain where the diffuser increases the flow velocity at the rotor plane.

Answer: Thanks for this comment. Unfortunately, we do not have generated a 3d version of this figure. We have recreated this figure in 2D plane and increased the quality to show more details of flow.

Round 2

Reviewer 3 Report

The manuscript has been nicely revised and I can recommend to be published in the current form.